# Immune Checkpoint Inhibitors: Changing the Treatment Landscape in Esophagogastric Adenocarcinoma

**DOI:** 10.3390/ph16010102

**Published:** 2023-01-10

**Authors:** Emer Lynch, Austin G. Duffy, Ronan J. Kelly

**Affiliations:** 1Department of Medical Oncology, The Mater Hospital, D18 DH50 Dublin, Ireland; 2The Charles A. Sammons Cancer Center, Baylor University Medical Center, Dallas, TX 75246, USA

**Keywords:** Gastric cancer, gastro-esophageal adenocarcinoma, immune checkpoint inhibitor, PD-L1, CTLA-4

## Abstract

In the West, recent decades have demonstrated an epidemiological trend towards esophago-gastric adenocarcinomas (EGAC), with considerable associated mortality. Historically, chemotherapy has represented the sole systemic treatment option in the advanced EGAC setting, in addition to complementing the role of surgery and radiotherapy in the case of localized disease. Immune checkpoint inhibitors (ICIs) represent a novel systemic therapeutic choice and have revolutionized the management of other malignancies, including melanoma and renal cell carcinomas. This article considers the rationale for ICIs in EGAC, reviews the evidence supporting their role in the current standard of care in EGAC, and briefly considers ongoing trials and future directions for the ICI class in EGAC.

## 1. Background

Adenocarcinoma of the esophago-gastric junction and lower esophagus, together with gastric adenocarcinoma, contribute significantly to cancer-related mortality and account for approximately one million deaths worldwide each year [1]. There are considerable regional variations in the incidence of gastroesophageal adenocarcinoma and, in recent decades, a striking change in the pattern in the US and Western populations where the predominance of cancer affecting these sites has been paralleled by a decrease in non-cardia/distal gastric cancer. During this time period the role of chemotherapy, both as a supplement to surgery in the curative setting, or the main modality of care for advanced disease, has become clearly established. Progress has been slow, however. Relapse rates following surgery are still too high and in the advanced disease setting the median survival times have not dramatically changed. New approaches are needed to both supplement the tri-modality standards used with curative intent and, for patients with advanced disease, to improve the duration of disease control. With the exception of Her2-directed treatments [2,3] the incorporation of more specific or targeting treatments into the management paradigm of gastroesophageal adenocarcinoma has been disappointing. The immune checkpoint inhibitor (ICI) class of drugs has had an enormous positive impact on the entire field of oncology over the past decade, radically altering the approach to cancers such as lung cancer or melanoma, by inducing durable and sometimes even complete responses in malignancies with previously limited prognoses. Gastrointestinal cancers have largely lagged behind in this regard, with the standard trimodality backbone of chemotherapy, radiotherapy and surgery remaining the standard of care in the early disease setting, and systemic chemotherapy in the advanced disease setting, until very recently. However, in the past couple of years this has changed, especially for esophago-gastric adenocarcinoma (EGAC) and in 2021 and 2022 in esophageal squamous cell carcinomas (ESCC). Here, we discuss the rationale for ICIs in EGAC and summarize recently published data for this approach in the perioperative and advanced disease setting. We also highlight important ongoing ICI trials that have the potential to change practice in the years ahead.

## 2. Rationale for an Immune Approach to EGAC

The immune system plays an under-appreciated role in the progression and evolution of cancer as well as being an important variable in how cancers respond to treatment (or not) [4,5]. The immune system contributes to the effectiveness of standard treatments such as chemotherapy and radiation (and may even be required for their efficacy [6]). The Cancer Genomic Atlas (TCGA) project classified gastric cancer into four distinct categories, and two of these: Epstein-Barr virus (EBV)-related and microsatellite instability (MSI) tumors, demonstrate activity in immune signalling pathways [7]. Almost 20% of gastric adenocarcinomas are MSI-high [8]. As we shall discuss, these subtypes predict for excellent responses to ICI. More broadly, the reactivity of the host immune system against cancer can be inferred by various pathologic findings, such as tumor infiltrating lymphocytes (TILs) embedded in resected tumors. The finding of TILs is evidence that an immune response is already in train, and this has been shown in a number of cancer types to correlate with prognosis. For example, in patients undergoing neoadjuvant chemotherapy for EGAC, which is treatment prior to surgery or other definitive intervention, TIL density was higher in responders and the density of lymphocytic invasion was an independent prognostic factor [9]. Conversely, the presence of inhibitory cells such as T regulatory cells has been shown to be a marker of poor prognosis [10]. Systemic and local therapies can impact the immune microenvironment, and is important to consider as we incorporate the ICI class into combination strategies with the more traditional modalities. The types and doses of chemotherapy and radiotherapy may be key variables. Certain chemotherapeutic agents (e.g., oxaliplatin) have more of a ‘pro-immune’ effect than others and this should be a particular consideration in trial design where peri-operative treatment is being administered [11]. The presence of TILs in a patient’s resected tumor is suggestive of an adjunctive role being played by the immune system, though clearly the immune response was inadequate to completely inhibit tumor growth. Alleviating the inhibition of the anti-tumor effector cell response is the underlying rationale for the ICI class which, by definition, works only when an anti-tumor reaction is already in motion. As our understanding of the role that the immune system plays in malignancies, including EGAC, has grown, treatment targets within the immune checkpoint pathway have been recognised. This has resulted in the development of new agents that target programmed cell death-ligand 1 (PD-L1), its receptor programmed cell death-1 (PD-1) and other immune checkpoints across the pathway. PD-L1 is expressed on both tumor cells and immune cells including macrophages, and is significant as both a predictive biomarker and as an indicator of immunogenicity. In EGAC, a greater level of CD8 T lymphocyte expression can be associated with higher PD-L1 expression, suggesting an adaptive immune response [12]. The PD-L1 combined positive score (CPS) is a clinically validated score generated by analysing the ratio of PD-L1 positive cells in comparison with all tumor cells and, as discussed below, is utilised as a as a predictive biomarker for ICI use in EGAC across a number of clinical trials [13].

## 3. Locoregional Disease

As shown in Table 1 a number of trials have evaluated or are currently evaluating ICIs in the locoregional setting. CHECKMATE 577 was a practice-changing phase III trial (*n* = 794 patients) that investigated the impact of adjuvant nivolumab, an anti-PD-1 monoclonal antibody, in patients with gastro-esophageal cancer [14]. All patients had resected stage II or stage III esophageal (adenocarcinoma or squamous cell histology) or gastroesophageal junction cancer, who had received combined modality neoadjuvant treatment and had residual disease (i.e., not a complete pathological response). The majority of patients in this study were male (84.5%) and had adenocarcinoma (71%). Disease-free survival was significantly longer in those that received nivolumab [22.4 months (95% confidence interval [CI], 16.6 to 34.0) vs. 11.0 months (95% CI, 8.3 to 14.3)]. Post hoc analyses showed that patients with tumors with a PD-L1 ≥ 5 by CPS had an improved disease-free survival (29.4 vs. 10.2 months). Readout on the secondary overall survival endpoint is expected in 2023, but based on the data to date both the United States Food and Drug Administration (FDA) and the European Medicines Association (EMA) have approved the use of adjuvant nivolumab for resected stage II/III esophageal/gastroesophageal junction (E/GEJ) cancers. Of note, as per historic CROSS data for neoadjuvant chemoradiotherapy in esophageal cancer, ESCC demonstrates greater pathological complete response (pCR) rates to chemoradiotherapy than EAC (esophageal adenocarcinoma) of 49% vs. 23% [15].

A number of studies looking at immunotherapy in the perioperative setting for gastric/GEJ cancers are currently underway. Interim results from DANTE were presented at the American Society of Clinical Oncology (ASCO) 2022 meeting [16]. A phase IIb randomized trial, it assesses pre- and post-operative FLOT +/− atezolizumab, an anti-PD-L1 antibody, followed by maintenance atezolizumab in the immunotherapy arm (arm A) in patients with ≥T2 tumors. In this study, 295 patients were randomized with a primary endpoint of progression free survival, and secondary endpoints of surgical outcome, pathological regression and safety. Chemotherapy completion rates were similar across both arms, with comparable surgical morbidity and mortality and R0 resection rates. The immunotherapy arm demonstrated better downsizing with a pT0 outcome of 23% in arm A and 15% in arm B. A pN0 resection was reached of 68% in arm A and 54% in arm B. Greater tumor regression was seen in arm A, with escalating levels of regression noted for higher PD-L1 scores. Centrally assessed pCR was seen in 46% of those with a PD-L1 CPS >/= 10 in arm A, in comparison with 24% in arm B. There were comparable pCR rates in allcomers, with approximately a quarter achieving pCR in both the experimental and the control arm. The improved efficacy in subgroups that are enriched for higher PD-L1 CPS status is consistent with the results of Checkmate 649, hence the EMA opinion to approve the use of nivolumab in the first line metastatic setting to only those patients with tumors exhibiting CPS ≥ 5.

A number of other studies in the peri-operative setting have not yet reported. EORTC VESTIGE, like CHECKMATE-577, evaluates post-operative ICI—in this case ipilimumab (an anti-CTLA-4 antibody) and nivolumab—in patients who did not have a pathological complete response following pre-operative treatment. It is a relatively large (*n* = 240) randomized phase 2 study, and unlike Checkmate-577 includes only patients with EGAC histology who are high risk with an R1 resection (resection margins are positive for malignancy) or positive lymph nodes post neoadjuvant chemotherapy, as per European and North American standard of care guidelines, followed by surgery. Patients will either receive the chemotherapy they had pre-operatively or the ICI combination (at dose and schedules of nivolumab 3 mg/kg IV q 2-weekly and Ipilimumab, 1 mg/kg IV q 6-weekly for 1 year) [18].

KEYNOTE-585 is a somewhat more traditionally designed phase III study of patients with previously untreated, localized, resectable gastric/GEJ adenocarcinoma in whom perioperative chemotherapy is indicated. The study randomizes patients to receive pembrolizumab or placebo in combination with chemotherapy [19]. Some have criticized the trial design, as only a small percentage have received the current standard of care chemotherapy backbone of FLOT. MATTERHORN is a randomized controlled phase III study investigating peri-operative FLOT +/− durvalumab, anti-PD-L1 monoclonal antibody [20]. Patients will receive either durvalumab 1500 mg or placebo every four weeks, with FLOT on days 1 and 15 of this 28-day cycle for a total of four cycles, two in the neoadjuvant setting and two in the adjuvant setting. This is followed by maintenance treatment with durvalumab or placebo every four weeks for a further ten cycles. The primary endpoint is event-free survival, and the secondary end points are overall survival and pCR rates, in addition to assessing tolerability and safety.

Finally, the role of immunotherapy has been firmly established in the advanced/metastatic setting in tumors that are microsatellite instability-high or mismatch repair deficient (MSI-H/dMMR). In 2017, the FDA granted its first tissue-agnostic approval for the use of pembrolizumab in the presence MSI-H/dMMR. Operable MSI-H gastric cancer has been identified as being less chemotherapy-sensitive, and so the use of neoadjuvant IO strategies to improve tumor downstaging is a rational choice requiring investigation. The NEONIPIGA Phase II study was recently reported, and in this trial the use of neoadjuvant ipilimumab (1 mg/kg every 6 weeks) with nivolumab 240 mg every 2 weeks for two cycles, followed by surgery then nivolumab 480 mg every four weeks for 9 cycles [17] was investigated. The primary endpoint was pCR rate. In total, 32 patients were enrolled, and all had complete endoscopic responses with tumor free biopsies. Of the three patients that did not proceed to surgery, two refused surgery, and one patient had metastasis at inclusion. Overall, 29 patients proceeded to surgery, all had an R0 resection and 58.6% had pCR. This trial continues to recruit as a phase II trial. While these early phase results are potentially practice-changing, in the absence of prospective randomized data it is as still difficult to move away from the perioperative standard of care, which is systemic chemotherapy with FLOT, as defined in current NCCN guidelines. On this note, there are a number of ongoing phase III clinical trials evaluating the use of immune checkpoint inhibitors in the early disease setting in gastro-esophageal adenocarcinoma; these are summarized in Table 1. Of note among them, EA2174 looks to assess perioperative nivolumab and ipilimumab in those treated with neoadjuvant chemoradiotherapy. Similarly, Keynote-975 aims to analyze the role of pembrolizumab with chemoradiotherapy in both esophageal adenocarcinoma and esophageal squamous cell carcinoma (ESCC). Furthermore, a host of phase III studies are currently underway focusing on immune checkpoint inhibitors in early-stage ESCC; however, these are outside the scope of the current review.

## 4. ICI for Advanced/Metastatic Disease

The first clinical trials suggesting efficacy for ICI in EGAC were for monotherapy in the advanced disease setting (Table 2). Pembrolizumab demonstrated encouraging response rates in gastric (KEYNOTE-012 [23]) and esophageal cancer (KEYNOTE-028 [24]) (overall response rate (ORR) 33% and 30% respectively). Subsequently, KEYNOTE-059 evaluated pembrolizumab in patients with metastatic gastric cancer who had received two previous lines of treatment [25]. The ORR was 16.4%, and was slightly higher for those whose tumors were PDL-1 greater or equal to one, i.e., 22.7% vs. 8.6% for PD-L1 negative. Based on these data, pembrolizumab was initially approved by the FDA in 2017, but this indication was subsequently withdrawn in 2021 following an FDA Oncologic Drug Advisory Committee recommendation with the development of data showing the value of ICI in the first line setting. Similarly, nivolumab, another monoclonal antibody blocking PD-1, first showed preliminary evidence of activity [26]. ATTRACTION-2 was a phase III placebo-controlled study randomizing 493 patients in heavily pre-treated Asian patients with gastric and junctional tumors. There was a modest improvement seen in median progression-free survival (PFS) and overall survival (1.61 vs. 1.45 months and 5.26 vs. 4.14 months, respectively) that favored the nivolumab arm, although it is important to point out that this, perhaps, more reflects the inadequacy of a median endpoint in a population where a small minority of patients were likely to respond. The landmark endpoints are perhaps more instructive with the 12-month OS rate in favor of nivolumab (26.6% vs. 10.9%). Interestingly, PD-L1 status in ATTRACTION-2 was not associated with benefit, with similar survival data being seen across PD-L1 positive and negative patients (5.22 and 6.05 months respectively).

Combined CTLA-4 and PD-1 blockade has a strong immunological rationale [40,41] in addition to being a standard option in cancer types such as kidney cancer or melanoma [42,43]. CHECKMATE-032 evaluated the combination of nivolumab/ipilimumab at different dose and schedules in *n* = 160 patients with advanced esophago-gastric cancer [37]. The addition of the anti-CTLA4 component did increase toxicity, but despite this the activity seen were grounds for further evaluation in the phase III CHECKMATE-649 study discussed below [44]. Another similar combination of tremelimumab, a fully humanised anti-CTLA-4 antibody, and durvalumab, was evaluated in a randomized phase II study in patients with advanced gastric/GEJ cancers. The 12-month overall survival rates for these combined immunotherapies were similar to those with ipilimumab and nivolumab [45].

KEYNOTE-181 evaluated pembrolizumab in a slightly earlier disease setting in patients who had received one previous line of chemotherapy [34]. Patients were assigned to either pembrolizumab or the investigator’s choice of chemotherapy (paclitaxel, irinotecan or docetaxel). This study included a majority of patients with squamous cell carcinoma (63.9%) with a total of 628 patients randomized. Overall treatment with pembrolizumab showed no overall survival benefit vs. chemotherapy alone, with median overall survival in both arms at 7.1 months in each group (HR, 0.89 (95% CI, 0.75 to 1.05); *p* = 0.0560). Pembrolizumab did improve overall survival in patients with a CPS ≥ 10, with a median OS of 9.3 months in the pembrolizumab arm in comparison with 6.7 months in the chemotherapy arm (HR 0.69 (95% CI, 0.52 to 0.93); *p* = 0.0074). Histopathological stratification also demonstrated improved median OS to 8.2 months in esophageal squamous cell carcinoma (ESCC), in comparison with the adenocarcinoma population; however, the study did not meet the coprimary end point. A caveat of interpreting these results is that the authors flag that the study was insufficiently powered to statistically analyze significant differences between patients with a PD-L1 ≥ 10 in either SCC or adenocarcinoma. These early studies were the first to indicate the importance of PD-L1 as a marker of efficacy in esophagogastric cancers, and that PD-L1 CPS was a better scoring system to use in adenocarcinoma histologies, whereas the tumor proportion score (TPS) was satisfactory for squamous carcinomas.

## 5. First Line Treatment with or without Chemotherapy

Prompted by the success of utilising single agent ICI in later line settings in KEYNOTE 181, in addition to benefits noted in other tumour types combining the ICI class with systemic chemotherapy, KEYNOTE 590 was undertaken. It was a randomised study in patients who had untreated advanced esophageal cancer and received either pembrolizumab with chemotherapy or chemotherapy alone [46]. The chemotherapy regimen for both arms consisted of cisplatin-5-fluorouracil administered every three weeks. Overall, while this study demonstrated improved survival for the addition of pembrolizumab to chemotherapy, this was probably driven by the ESCC patients who formed the majority (73%), especially those with a high PDL1 score (51%). Further updated results were presented at the 2022 ASCO Gastrointestinal Symposium. Median OS in ESCC was 12.6 months (HR 0.73, (0.61, 0.88)) in comparison with 11.6 months in the adenocarcinoma cohort (HR 0.73, (0.55–0.99)). The updated data demonstrated extended OS in both histopathological cohorts, which had not been seen with more immature data [47]. This highlights the potential for ICI therapy to deliver better durability of response in this cohort, working complementarily with chemotherapy, albeit for a minority of patients. The FDA approval for this regimen however is limited to esophageal (both ESCC and EAC) and siewert class 1 GEJ tumors. Keynote 590 did not enroll any gastric cancers, and at the time of writing pembrolizumab is not approved in this indication.

CHECKMATE-649 and ATTRACTION 4 both evaluated the combination of Nivolumab and chemotherapy for untreated metastatic esophagogastric adenocarcinoma [27,28]. CHECKMATE-649 evaluated nivolumab or ipilimumab/nivolumab in combination with oxaliplatin-based chemotherapy in patients who were HER2 negative (or unknown), irrespective of PD-L1 status [28]. During enrolment, the protocol was amended to focus on the evaluation of those with a PD-L1 CPS ≥ 5. This was based on the results of the gastro-esophageal cohort of CHECKMATE-032 in addition to other studies which suggested that PD-L1 CPS might be a better biomarker for anti-PD-1 therapy efficacy than TPS. However, patients continued to be enrolled irrespective of their PD-L1 status [13]. In total, *n* = 1581 patients were randomized, with approximately 60% having tumors expressing PD-L1 with CPS ≥ 5. Most (76%) patients were non-Asian and most (70%) had gastric cancer. The primary endpoints were OS and PFS in patients with PD-L1 CPS ≥ 5. Once the primary endpoints were met in this enriched population, it was possible to evaluate the secondary endpoints of OS in patients with PD-L1 ≥ 1 and in all randomized patients. The response rate in the CPS PD-L1 ≥ 5 population were higher in those patients who received nivolumab (45% vs. 60%, *p* < 0.0001). This translated into a median increase in PFS of 1.7 months and increase in OS of 3.3 months. As a result of these data, the US FDA approved Nivolumab in all comers, but giving category 1 evidence to those with gastro-esophageal adenocarcinoma with PD-L1 CPS ≥ 5 in comparison to category 2B evidence for those with PD-L1 CPS ≥ 1. In comparison, the EMA has only afforded regulatory approval in patients with CPS ≥ 5. At the European Society for Medical Oncology (ESMO) Congress in 2021, the Checkmate-649 data were updated to include the ipilimumab and nivolumab arm [48]. Whilst the addition of nivolumab to oxaliplatin-based chemotherapy continued to show improvement in survival compared to chemotherapy (14.4 months vs. 11 months) there was no survival advantage for the addition of combined blockade with ipilimumab and nivolumab in patients with PD-L1 CPS ≥ 5. It is therefore unclear if the addition of anti-CTLA4 blockade has a role in the evolving landscape for these tumor types. Additionally, the combination chemo-immunotherapy (single agent nivolumab with chemotherapy) arm did demonstrate increased toxicity in comparison with chemotherapy alone, with 59% of the combined arm experiencing grade 3–4 toxicity vs. 44% in the chemotherapy alone arm, which is not beyond what would be expected in combining treatment modalities.

ATTRACTION-4 was of a similar design to CHECKMATE 649, except it was primarily (or exclusively) in Asian patients and did not limit enrolment or assessment relative to PDL1 CPS [48]. Patients were randomized to oxaliplatin-based chemotherapy with or without nivolumab in advanced disease in the first line setting. The majority (90%) of patients had gastric adenocarcinoma. The addition of nivolumab improved PFS but not OS. At the prespecified interim analysis, in the nivolumab with chemotherapy arm PFS was 10.45 months (95% CI 8.44–14.75) in comparison with 8.34 months (95% CI 6.97–9.40) in the placebo plus chemotherapy group (HR 0.68 (98.51% CI 0.51–0.90); *p* = 0.0007). In the final analysis, median overall survival in the nivolumab with chemotherapy arm was 17.45 months vs. 17.15 months in the placebo plus chemotherapy group (HR 0.90; 95% CI 0.75–1.08; *p* = 0.26). At face value, overall survival data was negative with similar survival curve in both arms. However, two thirds of those enrolled in ATTRACTION-4 went on to receive further chemotherapy, with 27% in the chemotherapy arm receiving later immune checkpoint inhibitor therapy. This raises the possibility that later lines of therapy may have influenced overall survival data.

Orient-16 evaluated sintilimab, a fully human IgG4 monoclonal antibody targeting PD-1, in combination with oxaliplatin-based chemotherapy in the first-line treatment of gastric or junctional tumours [32]. There was a significant improvement in OS in the experimental arm in all patients (median 15.2 vs. 12.3 months), which was improved for those with PD-L1 CPS ≥ 5 (median 18.4 vs. 12.9 months). Table 3 summarizes ongoing clinical trials with immune checkpoint inhibitors in the advanced esophagogastric adenocarcinoma setting.

## 6. ICIs in Her 2-Postive EGAC

Up to 30% of esophago-gastric adenocarcinomas have ERBB-2 (or HER2) oncogenic amplifications, which have been successfully clinically targeted with a combination of systemic chemotherapy and anti-HER2 therapy in the form of trastuzumab for almost a decade, as demonstrated by the ToGA trial. Preclinical and clinical data had demonstrated that trastuzumab increases HER2 internalization and cross presentation by dendritic cells which ultimately stimulates HER2 specific T cell responses. Trastuzumab had also been shown to have significant effects on the immune microenvironment by upregulating PD-1 and PD-L1 expression, inducing TIL expression and modulating MHC class II expression. Therefore, combining an ICI with trastuzumab and chemotherapy in a HER2 positive population of patients was an intriguing prospect. The initial study was a small single arm, phase II study (*n* = 37) in which pembrolizumab was added to the combination of trastuzumab and chemotherapy in patients with untreated metastatic HER2-positive EGAC [58]. The results are very encouraging with 70% of patients progression-free at 6 months. The treatment was well tolerated with no concerning safety signals [59]. Initial data from the phase III KeyNote-811 study comparing pembrolizumab plus trastuzumab and chemotherapy to trastuzumab and chemotherapy alone were presented at the ASCO Annual meeting 2021 [60]. For the first 264 patients enrolled, the confirmed response rate (ORR) (95% CI) was 74.4% (66.2–81.6) vs. 51.9% (43.0–60.7) in favor of the pembrolizumab arm. A complete response rate of 11.3% was seen (compared to 3.1% in the standard treatment arm). Based on these preliminary results the FDA granted accelerated approval for this regimen.

Similarly, preliminary data from DESTINY-Gastric03 were presented at the ASCO GI Cancers Symposium 2022 [54]. A phase 1b/II open-label, dose-escalation and dose-expansion study, it is multi-armed, with the aim of investigating trastuzumab deruxtecan alone and in combination with a variation of chemotherapy +/− Durvalumab or Pembrolizumab in patients with pre-treated HER2+ advanced gastric cancer. Early results are promising with an ORR of 50%.

## 7. Other ICIs

Several other ICIs have entered the clinic in an attempt to build on the initial progress that has been made. Antibodies targeting various inhibitory molecules in the immune synapse include those binding lymphocyte-activation gene 3 (LAG-3), T-cell immunoglobulin and mucin containing-domain-3 (TIM-3) and T-cell immunoreceptor with immunoglobulin and immunoreceptor with tyrosine-based inhibitory motif domains (TIGIT) inhibitors. It is unclear, as yet, which of these compounds, if any, will find a niche. Anti-LAG-3 antibodies, such as relatimab [61], and anti-TIM-3 antibodies, sabatolimab [62] are being investigated in clinical trials as a monotherapy and in combination with other ICIs. Thus far, combination immune-based approaches have been largely disappointing in esophagogastric cancers, with basket studies such as FRACTION (BMS, NCT02935634) [63] and MORPHEUS (ROCHE, NCT03281369) [64] not resulting in an obvious candidate to bring forward to phase III. Perhaps the most eagerly awaited study is that combining anti-PD1 therapy with multikinase inhibition. The LEAP-015 Trial (NCT0466271) is a randomized phase III study evaluating pembrolizumab, lenvatinib and chemotherapy in patients with advanced, untreated EGAC. Lenvatinib is a multikinase inhibitor of vascular endothelial growth factor receptors (VEGFR) 1, 2 and 3, as well as fibroblast growth factor receptor (FGFR) and platelet-derived growth factor receptor (PDGFR), and has immune modulatory properties [65,66]. Safety run-in data were presented at the ESMO 2022 Congress, assessing the tolerability, safety, and efficacy of the combination. Overall, 15 patients were enrolled with an ORR of 73% demonstrated, and a disease control rate of 93%. Part 2 of the study, which is open-label and randomized, is currently enrolling.

## 8. Future Perspectives

The ICI drug class has dramatically altered the outlook and treatment paradigm of many cancers over the past decade. In the past two years, ICIs have demonstrated efficacy in EGAC and are now an integral part of the standard treatment paradigm for both operable and inoperable disease. There are, however, many questions, and the coming years will hopefully see a refinement of the approach. The role of PDL1 needs to be clarified or combined with other biomarkers to improve its predictive value. New techniques, such as circulating DNA in the post operative period, need to be married with evolving treatments to improve patient selection and maximize benefit. Newer immune modalities, either novel (or novelly combined) ICIs, cell-based approaches such as CAR-T, or tumor-infiltrating lymphocyte (TIL) therapies, offer hope of a radically different approach. The clinical studies summarized in this article have certainly established a role for ICI agents in this disease, and set a marker to be surpassed. Whilst newer drugs and strategies may achieve this independently, it is likely that the treatments will need to be integrated rather than supplanted. ICIs also present a new toxicity profile, that in combination with existing modalities demonstrate a not unexpected increase in adverse events. This is important to acknowledge moving forward, given it is unlikely that the standard of care chemotherapy and radiation dose and schedules we are currently using are optimal for anti-tumor immune effects. We can undoubtedly do a better job of incorporating immune-based approaches into the standard of care. This is especially challenging in the perioperative or first line metastatic setting. At the least, recent trials have shown that immune-based approaches have a role. We will, in time, discover how central this role is, and learn how to better select those most likely to derive benefit.

## Figures and Tables

**Table 1 pharmaceuticals-16-00102-t001:** Summary of relevant and awaited outcomes from immunotherapy trials in the locoregional esophagogastric setting.

Trial	Type	Line	Arms	DFS
CHECKMATE-577 [14]	Esophageal/GEJ cancers Global *n* = 532	Phase III Adjuvant (post NACT, with residual pathological disease)	A: Nivolumab B: Placebo	mDFS: 22.4 vs. 11.0 months, (A vs. B)
AIO-DANTE [16]	Resectable Gastric/GEJ Adenocarcinoma *n* = 295	Phase IIb Neoadjuvant/Adjuvant	A: FLOT + Atezolizumab B: FLOT	Downsizing pT0 23% A vs. 15% B pN0 68% vs. 54% pCR in PD-L1 CPS ≥ 10 46% vs. 24%
NEONIPIGA [17]	dMMR/MSI-H gastric/GEJ adenocarcinoma *n* = 32	Phase II Neoadjuvant/Adjuvant	Neoadjuvant Ipilimumab + Nivolumab then adjuvant nivolumab	58.6% pCR
VESTIGE [18]	Esophagogastric adenocarcinoma	Phase II Adjuvant (high risk resected disease)	A: Adjuvant chemo (as prior to surgery) B: NIVO3/IPI1	Recruiting
KEYNOTE-585 [19]	Gastric/GEJ cancer	Phase III Neoadjuvant/Adjuvant	A: Pembrolizumab + chemotherapy B: Placebo + chemotherapy	Recruiting
MATTERHORN [20]	Gastric/GEJ Adenocarcinoma	Phase III Neoadjuvant/Adjuvant	A: Durvalumab + FLOT B: FLOT	Recruiting
EA2174 [21] NCT03604991	Esophageal and GEJ Adenocarcinoma and ESCC	Phase III Perioperative nivolumab and ipilimumab with neoadjuvant chemoradiotherapy	A: Chemoradiotherapy (CROSS) B: CROSS + Nivo C: Nivolumab D: Ipilimumab + Nivolumab	Recruiting
Keynote-975 [22] NCT04210115	Esophageal and GEJ Adenocarcinoma and ESCC	Phase III Chemoradiotherapy +/− pembrolizumab	A: FP or FOLFOX chemotherapy with radiotherapy and 1 year pembrolizumab B: FP/FOLFOX + RT + placebo	Recruiting

**Table 2 pharmaceuticals-16-00102-t002:** Summary of relevant outcomes from immunotherapy trials in the advanced esophagogastric setting.

Trial	Type	Line	Arms	ORR	OS	PFS
ATTRACTION-4 [27]	Gastric/GEJ cancer Asian patients *n* = 724	Phase III First line metastatic	A: Nivolumab + chemo (SOX or CAPOX) B: Placebo + chemo	(57.5 vs. 47.8%; *p* = 0.0088) (A vs. B)	No statistically significant difference	10.5 vs. 8.3 mo (A vs. B) HR 0.68; 98.51% CI 0.51–0.90; *p* = 0.0007
CHECKMATE-649 [28]	Esophageal, GEJ, gastric adenoca Global *n* = 1581	Phase III First line metastatic	A: Nivolumab + Ipilimumab B: Nivo + chemo (CAPOX or FOLFOX) C: Chemo	In the CPS ≥ 5 response rates were higher in nivolumab-treated patients (45% vs. 60%, * p * < 0.00 fcroa01)	In the CPS ≥ 5; 3.3 month gain in OS (HR 0.71; 98.4% confidence interval (CI) 0.59–0.86; * p * < 0.0001)	In the CPS ≥ 5; 1.7 month gain in PFS [hazard ratio (HR) 0.68 (0.56–0.81); * p * < 0.0001]
KEYNOTE-590 [29]	Esophageal, GEJ cancer Global *n* = 749	Phase III First line metastatic	A: Pembrolizumab + chemo (Cisplatin + 5-FU) B: Placebo + chemo	45.0% vs. 29.3% (*p* < 0.0001) in all pts, (A vs. B)	ESCC CPS ≥ 10; 13.9 vs. 8.8 mo ESCC; 12.6 vs. 9.8 mo; CPS ≥ 10; 13.5 vs. 9.4 mo All pts; 12.4 vs. 9.8 mo; (A vs. B)	ESCC; 6.3 vs. 5.8 mo; CPS ≥ 10; 7.5 vs. 5.5 mo All pts; 6.3 vs. 5.8 mo (A vs. B)
KEYNOTE 062 [30]	Gastric/GEJ cancer Global *n* = 763	Phase III First line metastatic	A: Pembrolizumab B: Pembrolizumab + chemo (Cisplatin + 5-FU or Capecitabine) C: Chemo	-	Pembrolizumab was not inferior to chemotherapy CPS of 1 or greater (median, 10.6 vs. 11.1 months; hazard ratio (HR), 0.91; 99.2% CI, 0.69–1.18) Pembrolizumab monotherapy was not superior to chemotherapy in patients with CPS of 1 or greater Pembrolizumab prolonged OS vs. chemotherapy in patients with CPS of 10 or greater (median, 17.4 vs. 10.8 months; HR, 0.69; 95% CI, 0.49–0.97)	CPS of 1 or greater (6.9 vs. 6.4 months; HR, 0.84; 95% CI, 0.70–1.02; *p* = 0.04
KEYNOTE 059 [31]	Gastric/GEJ adeno Global = 56	Phase II First line metastatic	A: Pembrolizumab and Cisplatin/5-FU B: Pembrolizumab	A: 60.0% [95% confidence interval (CI), 38.7–78.9] B: 25.8% (95% CI 11.9–44.6)	-	-
ORIENT-16 [32]	Gastric/GEJ adeno *n* = 650	Phase III First line metastatic	A: Sintilimab and chemo (CAPOX) B: Placebo and chemo		CPS ≥ 5 (median 18.4 vs. 12.9 mo; HR 0.660; 95% CI 0.505–0.864; *p* = 0.0023) All pts (median 15.2 vs. 12.3 mo; HR 0.766; 95% CI 0.626–0.936; *p* = 0.0090) (A vs. B)	CPS ≥ 5 (HR 0.628; 95% CI 0.489–0.805; *p* = 0.0002) All pts (HR 0.636; 95% CI 0.525–0.771; *p* < 0.0001) (A vs. B)
Javelin Gastric 100 [33]	Gastric/GEJ cancer Global *n* = 805	Phase III First line maintenance metastatic	A: Avelumab B: Chemo (continue oxaliplatin and 5-FU)	-	mOS was 10.4 vs. 10.9 mo 24-month OS rate was 22.1% vs. 15.5% PD-L1-positive population the HR for OS was 1.13 CPS ≥ 1 mOS was 14.9 months vs. 11.6 mo (A vs. B)	-
KEYNOTE-181 [34]	Esophageal cancer *n* = 628	Second line metastatic	A: Pembrolizumab B: Chemotherapy (investigators choice)	-	CPS ≥ 10; 9.3 vs. 6.7 months; hazard ratio [HR], 0.69 [95% CI, 0.52 to 0.93]; *p* = 0.0074) ESCC: Median OS was 8.2 months vs. 7.1 months (HR, 0.78 (95% CI, 0.63 to 0.96); *p* = 0.0095) All patients: 7.1 months vs. 7.1 months (HR, 0.89 (95% CI, 0.75 to 1.05); *p* = 0.0560) (A vs. B)	-
KEYNOTE 061 [35]	Gastric/GEJ cancer Global *n* = 592	Phase III Second line metastatic	A: Pembrolizumab B: Paclitaxel	-	9.1 months (95% CI 6.2–10.7) with pembrolizumab and 8.3 months (7.6–9.0) with paclitaxel (hazard ratio (HR) 0.82, 95% CI 0.66–1.03; one-sided *p* = 0.0421	Median PFS 1.5 months (95% CI 1.4–2.0) with pembrolizumab and 4.1 months (3.1–4.2) with paclitaxel (HR 1.27, 95% CI 1.03–1.57)
Attraction 2 [36]	Gastric/GEJ cancer Asian *n* = 493	Phase III Refractory disease	A: Nivolumab B: Placebo	-	OS was significantly longer in the nivolumab group (5.26 (4.60–6.37) vs. 4.14 (3.42–4.86) months in placebo group) at the 2-year follow-up	-
Checkmate 032 [37]	Esophageal, GEJ, gastric cancer US/Europe *n* = 160	Phase II Refractory disease	A: NIVO3 B: NIVO1/IPI3 C: NIVO3/IPI1	A: 12% B: 24% C: 8%	12-month OS rates A: 39% B: 35% C: 24%	12-month PFS rates A: 8% B: 17% C: 10%
KEYNOTE180 [38]	Esophageal cancer Global *n* = 121	Phase II Refractory disease	A: Pembrolizumab Single arm	All: 9.9% ESCC: 14.3% Adeno: 5.2% PDL1 pos: 13.8% PDL1 neg: 6.3%	-	-
Javelin Gastric 300 [39]	Gastric/GEJ cancer Global *n* = 371	Phase III Refractory disease	A: Avelumab B: Chemotherapy (investigators choice)	Did not meet secondary end points of imoriving ORR (2.2% vs. 4.3%) (A vs. B)	Did not meet primary end point of improving OS (median, 4.6 vs. 5.0 months) (A vs. B)	Did not meet secondary end point of improving PFS (median, 1.4 vs. 2.7 months) (A vs. B)

**Table 3 pharmaceuticals-16-00102-t003:** Summary of ongoing clinical trials with immune checkpoint inhibitors in the advanced esophagogastric adenocarcinoma setting.

Trial	Type	Line	Arms	Status
KEYNOTE-859 [49]	Phase III Gastric/GEJ Adenocarcinoma	First Line	A: Pembrolizumab + Chemotherapy (CAPOX or 5 Fu + Cisplatin) B: Chemotherapy	Active, not recruiting
NCT03745170 [50]	Phase III Gastric/GEJ Adenocarcinoma	First Line	A: CAPOX + Sintilimab B: CAPOX	Active, not recruiting
NCT03777657 [51]	Phase III Gastric/GEJ Adenocarcinoma	First Line	A: Chemo (CAPOX or 5 Fu + Cisplatin) + Tislelizumab B: Chemo + placebo	Active, not recruiting
NCT05144854 [52]	Phase III Gastric/GEJ Adenocarcinoma	Chemotherapy naïve	A: Ipilimumab + Nivolumab + Chemotherapy (CAPOX/SOX) B: Chemotherapy	Recruiting
KEYNOTE-811 [53] NCT03615326	Phase III HER2+ Gastric/GEJ Adenocarcinoma	First Line	A: Pembrolizumab + Trastuzumab + Chemo (CAPOX or 5 Fu Cisplatin or SOX) B: Trastuzumab + chemotherapy	Active, not recruiting Interim analysis: Arm A: ORR 74.4% (95% CI, 66.2–81.6) Arm B: 51.9% (95% CI, 43.0–60.7) 22.7% improvement in ORR in the pembrolizumab group (95% CI, 11.2–33.7; *p* = 0.00006)
DESTINY-Gastric03 [54]	Phase Ib/2 HER2+ Gastroesophageal/GEJ adenocarcinoma	Part 1: 2nd Line + Part 2: First Line	1a. Trastuzumab Deruxtecan (T-DXd) + 5 Fu 1b. T-DXd + capecitabine 1c. T-DXd + durvalumab 1d. T-DXd + 5 Fu or capecitabine + Oxaliplatin 1e. T-DXd + 5 Fu or capecitabine + durvalumab 2a. Traztuzumab + 5 Fu or capecitabine + Oxaliplatin 2b. T-DXd 2c. TDXd + 5 Fu or capecitabine + oxaliplatin 2d. T-DXd, 5 Fu or capecitabine + pembrolizumab 2e. T-DXd + pembrolizumab	Recruiting
HERIZON-GEA-01 [55] NCT05152147	Phase III HER2+ Gastric/GE Adenocarcinoma	First Line	A: Trastuzumab + chemotherapy (CAPOX or 5 Fu + Cisplatin) B: Zanidatamab (novel bispecific anti-HER2 +) + chemotherapy C: Zanidatamab + tislelizumab (anti-PD-1) + chemotherapy	Recruiting
INTEGRATEIIb [56] NCT04879368	Phase III Gastro-esophageal adenocarcinoma (or undifferentiated)	3rd line +	A: Nivolumab + Regorafenib B: SOC chemotherapy	Recruiting
LEAP-015 [57]	Phase III Gastroesophageal Adenocarcinoma	First Line	A: Lenvatinib + Pembrolizumab + Chemotherapy (CAPOX or mFOLFOX) B: Chemotherapy	Recruiting

## Data Availability

Not applicable.

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
