# Peer review of "Immune Checkpoint Inhibitors: Changing the Treatment Landscape in Esophagogastric Adenocarcinoma"

_pharmaceuticals, 2023, doi:10.3390/ph16010102_

Round 1

Reviewer 1 Report

Dear authors, the review "Immune checkpoint inhibitors: changing the treatment landscape in gastric cancer" is very relevant in the area of study. The manuscript possesses pertinent objectives and good results. However, I consider that some points should be improved.

1. Table 1 should contain a reference for each clinical trial. A more detailed comment on the manuscript is expected for each  Immune checkpoint inhibitor reaching clinical trials.

2. Is there information about the toxicity of Immune checkpoint inhibitors? This data will be relevant, mainly in vivo.

3. How many articles do the authors study to compose this review?

4. Kindly correlate the relationship between Immune checkpoint inhibitors and gastric cancer in clinical trials.

5. I suggest making it clear at what point it is really relevant to search for new Immune checkpoint inhibitors since there is a large number of molecules in ongoing clinical trials.

6. For a good review paper, figures are very important, authors need to add at least 3-5 figures either drawing them or taking them from recent relevant literature. 

7. The title of this manuscript contains the challenges term, but I didn't notice any clear challenges. It would be better if authors add one separate section and add challenges. 

8.  Authors focused on immune checkpoint inhibitors, but what are the distinguished properties and specific problems of immune checkpoint inhibitors? The authors never discussed it.

9.  According to the applications, most, if not all are applicable to other kinds of cancers. Then why did the authors not expand the topic to other cancer therapy? 

10. This manuscript is well organized but lacks specific comparative analysis. What are the advantages of "immune checkpoint inhibitors" compared with drugs for gastric cancer therapy?

11.  Future perspective is also missing kindly write it in a separate section. 

12.  In conclusion, the author should consider giving some methodological design about how to improve the performance of immune checkpoint inhibitors in the future.

13. Please revisit the entire manuscript for minor typo issues.

Author Response

Please see the attached word document. 

Many thanks.

Author Response

Many thanks for your kind review, please see the attachment. 

Reviewer 3 Report

Background- The immune checkpoint inhibitor (ICI) class of drugs have had an enormous impact on the entire field of oncology over the past decade, radically altering the approach to cancers such as lung or melanoma. Gastrointestinal cancers have largely lagged behind in this but in the past couple of years this has changed, especially for esophago-gastric adenocarcinoma (EGAC) and in 2021 and 2022 in esophageal squamous cell carcinomas (ESCC). The authors need to first state the enormous (positive) impact of ICI on the entire field of oncology. Second, they need to state why GI cancers have lagged behind.

Rationale- Most would argue that tumor immunology is now widely appreciated. Second, please state why ICIs should be considered for GI cancers besides the failure of prior regimens. Is there a molecular or cellular premise to support the use of ICIs.

Locoregional control- The phase 3 CHECKMATE 577 trial investigated the impact of adjuvant nivolumab in patients with esophageal or GE junction cancer. Why was greater benefit of nivolumab than placebo seen in patients with squamous-cell cancer, than for those with adenocarcinoma, node-negative disease and node-positive disease, and across tumor stages.

What is the advantage of Vestige over Keynote 577?

What is the rationale for Keynote 590?

pg. 5 - During enrolment, the protocol was was amended to focus--- spelling and grammar.

What are the putative advantages of sintilimab?

The putative benefit of the trials listed in table 2 should be addressed.

The basis for the trials listed in table 3 should be included.

Ref. 7- Citation omitted the authors/ network.

Author Response

(The authors gave the same response as above.)

Round 2

Reviewer 1 Report

Accepted in present form 

Author Response

Many thanks for taking the time to review and for accepting our article in its current form. 
